# Modelling the Impacts of <u>Historical and Future</u> Extreme Precipitation <u>Days</u> on <u>Seasonal</u> Surface Mass Balance in the Eastern Canadian

## **Arctic and Greenland**

Nicole A. Loeb<sup>1</sup>, Alex Crawford<sup>1</sup>, Brice Noël<sup>2</sup>, Julienne Stroeve<sup>1,3,4</sup>

- 5 Department of Environment & Geography and Centre for Earth Observation Science, University of Manitoba, Winnipeg, MB, Canada
  - <sup>2</sup> Laboratory of Climatology, Department of Geography, SPHERES research unit, University of Liège, Liège, Belgium,
  - <sup>3</sup> Department of Earth Sciences, University College London, London, United Kingdom
- <sup>4</sup> National Snow and Ice Data Center, Cooperative Institute for Research in Environmental Sciences, University of Colorado Boulder, Boulder, CO, United States

Correspondence to: Nicole A. Loeb (loebn@myumanitoba.ca)

Abstract. Land ice in the Arctic is losing mass as temperatures increase, contributing to global sea level rise. While this loss is largely driven by melt induced by atmospheric warming, precipitation can alter the rate at which loss occurs depending on its intensity and phase. Case studies have illustrated varied potential impacts of extreme precipitation events on the surface mass balance (SMB) of land ice, but the importance of extreme precipitation to seasonal SMB has not been investigated. In this study, simulations from the Regional Atmospheric Climate Model (RACMO) and Variable-Resolution Community Earth System Model (VR-CESM) are explored over historical (1980-1998) and future (2080-2098, SSP5-8.5) periods to reconstruct and further project seasonal SMB for the Greenland Ice Sheet and ice caps of the Eastern Canadian Arctic. Historically, extreme precipitation days consistently had higher SMB than non-extreme precipitation days throughout the study area in both the cold season (DJFM) and warm season (JJAS). In future simulations, this relationship persists for the cold season. However, for the warm season, projections indicate a shift towards less positive and more variable SMB responses to extreme precipitation days in the future, accounting for a larger portion of cumulative seasonal positive and negative SMB. Mass loss during extreme precipitation days becomes more common, particularly in SW Greenland and Baffin Island. This likely occurs in part because of a shift toward more rainfall during extreme precipitation events. In other words, in a strong warming scenario, extreme warm season precipitation and surrounding ice caps.

#### 1 Introduction

Arctic land ice, including the Greenland Ice Sheet (GrIS) and glacier and ice caps of the eastern Canadian Arctic, has been losing mass at an accelerated rate as the climate has warmed (e.g., Hugonnet et al. 2021; Constable et al. 2022). This mass loss is contributing to global sea level rise (e.g., Bamber et al., 2018; Hofer et al., 2020; Jacob et al., 2012) and triggers further warming via the ice-albedo feedback (e.g., Ryan et al., 2023). The ice-albedo feedback is one of the main drivers of "Arctic amplification", which refers to the Arctic region warming up to four times faster than the global average (Rantanen et al., 2022), in turn enhancing the rate of ice loss. The GrIS has been one of the largest contributors to global sea level rise since 1900 (van den Broeke et al., 2016; Fettweis et al., 2013; Frederikse et al., 2020; Hofer et al., 2020). A key driver of Greenland's

Deleted: Events

**Deleted:** Laboratoire de Climatologie et Topoclimatologie, SPHERES, University of Liège, Liège, Belgium

Deleted: and extreme precipitation events

Deleted: will

Deleted: is

**Deleted:** approximately

Deleted: Greenland Ice Sheet (

Deleted: )

1

contribution to global sea level rise is increased surface ice melt and runoff (e.g., Box, 2013; Fettweis et al., 2017). Annual and seasonal surface mass balance (SMB) of the GrIS has been extensively studied through observations (e.g., Bolch et al., 2013; Box, 2013; Cogley, 2004) and modelling (e.g., van Kampenhout et al., 2020; Noël et al., 2018a). The smaller ice caps and glaciers in the eastern Canadian Arctic Archipelago (CAA) have experienced accelerated mass loss in recent decades (Noël et al., 2018b). Lenaerts et al. (2013) showed that 18% of the land ice in the eastern CAA may be lost by 2100, even under a moderate warming scenario.

In models, the SMB is often quantified as

50

$$SMB = PR - RU - SU - ER \tag{1}$$

where *PR* refers to precipitation, *RU* is runoff, *SU* is loss due to sublimation/phase change, and *ER* represents wind-driven erosion (Noël et al., 2017, 2018b). The SMB neglects dynamic processes leading to ice loss, such as calving. In general, precipitation is expected to increase in most glaciated regions due to increased water vapour holding capacity as a result of atmospheric warming (e.g., Bengtsson et al., 2011; Norris et al., 2019; Skific et al., 2009). Surface melt has historically been the dominant factor driving land ice mass loss across much of the Arctic, largely due to rapid temperature increases and relatively low interannual variability in precipitation (Koerner, 2005; Van As et al., 2014). However, as the climate continues to warm, precipitation variability is expected to increase (Pendergrass et al., 2017), suggesting that precipitation may have a more critical impact on the variability of SMB in the future.

The SMB response to precipitation may change as the structure of the firn layer evolves with atmospheric warming. Firn is made up of snow that has lasted at least one melt season but has not yet compacted into glacial ice (Cogley et al., 2011). It is important when considering melt water and liquid precipitation, as it contains interconnected pore spaces that allow for liquid infiltration and freezing/refreezing, resulting in internal accumulation and reducing the amount of mass lost during melt (Forster et al., 2014; van Pelt and Kohler, 2015). However, the firn pore space is limited, and less may be available for retention as more melt and liquid precipitation occur (Machguth et al., 2016; Noël et al., 2022; van Pelt and Kohler, 2015). Noël et al. (2018b) noted how glaciers in the southern CAA are already experiencing decreased refreezing due to the filling of pore spaces, which has also been observed on the GrIS (MacFerrin et al., 2019). In addition to filling firn pore space, intense rainfall events can cause the densification of existing firn and prevent further firn growth (Machguth et al., 2016; Noël et al., 2017), meaning that more surface mass loss may occur due to rainfall in the future.

Another important factor when considering how precipitation may affect SMB is the rate of precipitation. Historical case studies have illustrated how extreme precipitation events can have different impacts depending on the timing and phase of precipitation. During the warm season, intense rainfall events have been shown to dramatically increase runoff and ice discharge (e.g., Doyle et al., 2015), causing the development of ice lenses that prevent infiltration and (re)freezing of liquid

Deleted: due to

Deleted: following

Deleted: e

water in firm (e.g., Box et al., 2022). Increased surface melt warms the firm as refreezing releases latent heat at depth during infiltration (e.g., Harper et al., 2023). Doyle et al. (2015) examined rainfall associated with a late summer extratropical cyclone over Western Greenland. The Kangerlussuaq region received approximately 20% of its annual precipitation in a period of seven days, which is very uncommon for the area. This caused a dramatic increase in melt water runoff and acceleration of ice flow. While the cyclone brought warmer temperatures that promoted surface melt, latent heat was released as the rainfall froze to the ice surface, and surface albedo decreased. This caused melt production well into the accumulation region of impacted glaciers. Conversely, a heavy snowfall event can increase the albedo and reduce summer melt (e.g., Noël et al., 2015). Bailey & Hubbard (2025) presented an analysis of a March 2022 atmospheric river event impacting the southeastern GrIS. Temperatures remained near 0°C, resulting in heavy snowfall across the region. The fresh snowfall increased surface albedo, delaying the onset of the melt season by 11 days. The effect of the reduced melt and added snow mass from the event was found to offset seasonal mass loss by approximately 8% during the following melt season, While extreme precipitation events can cause dramatic short-term SMB changes, either positive or negative, their importance in a seasonal context has not been explicitly studied.

Climate model simulations project that extreme precipitation events will shift in the future. While mean precipitation is slowly changing, observations have shown that precipitation extremes have shifted more quickly than mean conditions (Fischer and Knutti, 2016; Myhre et al., 2019; Pendergrass et al., 2017). Loeb et al. (2024) showed how extreme precipitation increases across much of the Baffin Bay and Greenland region in simulations of warming scenarios in the Variable-Resolution Community Earth System Model (VR-CESM). Climate model simulations project that a higher portion of annual precipitation will originate from extreme events. One of the factors driving this increase is atmospheric rivers occurring farther north than historically observed (Li & Ding, 2024; Loeb et al., 2024), which can bring high temperatures and extreme precipitation (e.g., Bao et al., 2006; Browning and Pardoe, 1973; Mattingly et al., 2018). Conversely, southeastern Greenland is projected to experience a decrease in the amount of extreme precipitation, likely related to reduced cyclone frequency and intensity in the region (Crawford et al., 2023; Loeb et al., 2024; Priestley and Catto, 2022).

105

Changing precipitation extremes will impact the rate at which mass loss occurs from the GrIS and ice caps of the eastern Canadian Arctic and therefore may accelerate or decelerate their contributions to sea level rise. While case studies have illustrated the complex impacts of individual extreme precipitation events on the short-term SMB of land ice, the overall importance of extreme events at seasonal time scales has not been investigated. In this study, two climate models are used to investigate the contributions of extreme precipitation events to seasonal SMB of the GrIS and neighbouring ice caps of the eastern Canadian Arctic, and how those contributions differ between historical simulations and climate projections under a high emissions scenario.

Deleted: during the warm season

**Deleted:** Oerlemans & Klok (2004) presented observations of a summer snowfall event in the Swiss Alps. An extratropical cyclone caused temperatures to fall by approximately 15°C and a zone of heavy snowfall impacted parts of the Alps for several days. The fresh snowfall led to increased albedo and reduced melt for several days following the event, even when temperatures increased.

Deleted: and annual

#### 2 Data & Methodology

#### 2.1 Model Simulations

#### 2.1.1 Regional Atmospheric Climate Model (RACMO)

The polar version of the Regional Atmospheric Climate Model (RACMO; van Meijgaard et al., 2008) is widely used to investigate the SMB of polar ice sheets (e.g., Lenaerts et al., 2013; Noël et al., 2017, 2018a). It contains a multi-layer snow module (40 layers) that reproduces processes within the snow column, including melt, percolation, refreezing, and runoff (Ettema et al., 2010). The amount of liquid water retention by capillary forces, or irreducible water saturation threshold, is set to 2% in RACMO2.3p2 (Glaude et al., 2024). Parameterization of snow surface albedo is based on prognostic snow-grain size, solar zenith angle, cloud optical thickness, and snow impurities (Kuipers Munneke et al., 2011).

130

The simulation used here is that of Noël et al. (2020, 2021); RACMO version 2.3p2 is used to dynamically downscale a Coupled Model Intercomparison Project (CMIP6) historical simulation of the Community Earth System Model (CESM) in 1950-2014, followed by a simulation of the SSP5-8.5 scenario in 2015-2100 with a spatial resolution of 11 km. Forcing of atmospheric temperature, pressure, specific humidity, wind speed and direction, sea ice, and sea surface temperature are prescribed at 6-hourly intervals (Noël et al., 2020, 2021).

## 2.1.2 Variable-resolution Community Earth System Model (VR-CESM)

The National Center for Atmospheric Research's Community Earth System Model (CESM), version 2.2, is a global earth system model that contains component models for the atmosphere, land, ocean, and cryospheric systems (Danabasoglu et al., 2020). The default spatial resolution of CESM is  $1^{\circ} \times 1^{\circ}$  latitude-longitude (Danabasoglu et al., 2020), but variable-resolution grids have been developed to downscale CESM simulations over areas of interest (Herrington et al., 2022). The Arctic VR-CESM grid is refined to  $0.25^{\circ} \times 0.25^{\circ}$  latitude-longitude over the entire Arctic nested within the  $1^{\circ} \times 1^{\circ}$  global simulation (Herrington et al., 2022).

The land component, the Community Land Model, version 5 (CLM5), simulates hydrological and snow processes, including SMB components for grid cells containing land ice (Danabasoglu et al., 2020; Lawrence et al., 2019). To account for the complex topography in glaciated areas, each glaciated grid cell is divided into 10 elevation classes to adjust atmospheric surface temperature, potential temperature, specific humidity, density, and pressure over ice surfaces (Lawrence et al., 2019). Along the periphery of ice caps and the GrIS, grid cells are also sub-divided into different land types to account for surface heterogeneity. CLM5 also redistributes precipitation produced by the atmospheric component model, the Community O Atmosphere Model, version 6 (CAM6) over glaciers. Precipitation is assumed to be snow below -2°C and rainfall above 0°C, with a mix occurring between the two thresholds (Lawrence et al., 2019).

**Deleted:** The snow cover is modelled with up to 12 layers and may reach a depth of 10 m water equivalent (w.e.) (Lawrence et al., 2019).

The SMB in CLM5 is calculated as in Eq. 1, except that ER is not explicitly modelled and is therefore not considered (van Kampenhout et al., 2020). Melt is determined based on the surface energy balance calculated over the top few centimeters of snow or ice (van Kampenhout et al., 2020). The snow model within CLM5 contains up to 12 layers, representing up to 10 m water equivalent (w.e.) of firn or snow (van Kampenhout et al., 2017; Lawrence et al., 2019). This allows for representation of processes such as compaction and liquid water percolation and retention within the column, with an irreducible water saturation threshold of 3.3% in CLM5 (van Kampenhout et al., 2020). Further details of the calculation of SMB in CLM5 are provided in van Kampenhout et al. (2020). The downscaling of CLM5 within VR-CESM has been shown to improve precipitation rates in the Arctic (Herrington et al., 2022; Loeb et al., 2024) and SMB of the GrIS (van Kampenhout et al., 2019).

65 Historical (1 Jan 1980 - 31 Dec 1998; Herrington et al., 2022) and future (1 Jan 2080 - 31 Dec 2098; Loeb et al., 2024) simulations were completed following the procedure of the Atmospheric Model Intercomparison Project (Hurrell et al., 2008), where the land (CLM5) and atmosphere (CAM6) components are actively modelled and coupled and sea surface temperatures and sea ice conditions are prescribed monthly. Monthly sea ice and sea surface temperatures are retrieved from existing CESM CMIP6 simulations (Danabasoglu et al., 2020; Meehl et al., 2020). The future simulation follows SSP5-8.5.

#### 170 2.2 Methods

The study domain is divided into nine subregions (Fig. 1); Canadian subregions are split by island. Greenland is divided into six regions based on glacier regime and SMB characteristics (Rignot et al., 2011; Rignot & Mouginot, 2012). The historical period (HIST) used is 1980-1998 and the future period (FUT) is 2080-2098, limited by the availability of VR-CESM data. Mean annual temperature in the study region rises in FUT relative to HIST by 6.1°C and 7.4°C in RACMO and VR-CESM, respectively. Two seasons are included for analysis: the warm season (JJAS) and cold season (DJFM). Four-month seasons are used, rather than three, to increase the number of extreme precipitation days that can be included for analysis and increase signal-to-noise ratio.

Deleted: -

Deleted:

**Deleted:** (Figure 1. Study domain map showing subregions used for analysis.)

Figure 1. Study domain map showing subregions used for analysis.

Extreme precipitation is defined in two ways for this study: by individual grid cell, which highlights spatial gradients in extreme precipitation and its impacts, and by subregion, which facilitates the analysis of events that are extreme over an entire drainage basin or island. Extreme precipitation days in each grid cell (mm w.e./day) are those for which total daily precipitation is at or above the 95<sup>th</sup> percentile of days with at least 1 mm of precipitation, following Loeb et al. (2022, 2024). At the subregion level, extreme precipitation days are defined as the days at or above the 95<sup>th</sup> percentile of total daily precipitation volume (m³/day) in the subregion, calculated by summing the product of precipitation multiplied by grid cell area for all glaciated grid cells in a subregion. To compare SMB on extreme precipitation days to non-extreme days, non-extreme precipitation days are defined as days where at least half of a region's grid cells receive at least 1 mm of precipitation, but the total amount is less than the extreme threshold for the subregion. In both cases, the historical threshold is used for both periods to assess changes in impacts resulting from precipitation at or above the same threshold. Historical extreme precipitation accumulations are compared to the 5<sup>th</sup> generation reanalysis product from the European Centre for Medium-Range Weather Forecasts (ERA5; Hersbach et al., 2020) to contextualize historical performance of RACMO and VR-CESM, following Loeb et al. (2024).

Short term anomalies in SMB-related variables for each extreme precipitation day were calculated relative to a window of ±15 days centred on the extreme precipitation day. We selected a 31-day period as the baseline instead of a climatology to focus on the within-season anomaly. This removes effects of background changes in mean seasonal/annual SMB conditions but will underestimate anomalies when the extreme events' impacts on SMB variables last for several days, which is likely most common for the albedo anomalies (e.g., Bailey and Hubbard, 2025; Oerlemans and Klok, 2004).

#### Deleted:

**Deleted:** to allow for both spatial (grid cell by grid cell) and regional analysis across the subregions of the domai

Deleted: n.

Deleted: over all grid cells

Deleted: and
Deleted: overall

Deleted: glaciated

Formatted: Superscript

**Deleted:** forintegrating the total quantity of precipitation over all glaciated grid cells

## Deleted:

Formatted: Not Highlight

**Deleted:**, leading to slight reductions in the magnitude of anomalies....

Next, the difference between historical and future (FUT minus HIST) interquartile range (IQR<sub>diff</sub>) of SMB anomalies on extreme precipitation days was calculated. The IQR represents the difference between the first quartile (25th percentile) and third quartile (75th percentile) of the data. To assess statistical significance of this difference, a bootstrapping method was employed in which all years were randomly sorted into two groups and the  $IQR_{diff}$  was calculated. Repetitions were performed 1000 times, and if the real  $IQR_{diff}$  was greater than (respectively less than) 975 of the tests, this indicated a statistically significant increase (respectively decrease) in IOR in the future simulation, yielding a two-tailed confidence interval of 95%. Note that some of the anomalies from VR-CESM are presented in the supplementary information.

To assess the relative importance of extreme precipitation days to seasonal SMB, we first grouped each day (i) of SMB in 225 each season into positive SMB  $(SMB_i^+)$  or negative SMB  $(SMB_i^-)$ . Second, we calculated the cumulative positive  $(SMB_{all}^+)$ and negative SMB  $(SMB_{all}^{-})$  during a season:

$$SMB_{all}^{+} = \sum SMB_{i}^{+} \tag{2}$$

$$SMB_{all}^{-} = \sum SMB_i^{-}$$
 (3)

Third, the same metric was calculated only including extreme precipitation days with positive (negative) SMB for  $SMB_{ex}^{+}$  $(SMB_{ex}^{-})$ . Finally, the mean fraction of seasonal positive and negative SMB was calculated as

$$SMB_{ex\ frac}^{+} = \frac{SMB_{ex}^{+}}{SMB_{all}^{+}}$$

$$SMB_{ex\ frac}^{-} = \frac{SMB_{ex}^{-}}{SMB_{all}^{-}}$$
(5)

$$SMB_{ex\,frac}^{-} = \frac{SMB_{ex}^{-}}{SMB_{ell}^{-}} \tag{5}$$

## 3 Extreme precipitation

To understand the impacts of extreme precipitation on SMB, we first investigate the occurrence of extreme precipitation and its seasonal and long-term changes. The mean monthly extreme precipitation accumulation in each subregion is shown in Fig. 2 to illustrate historical and future conditions across the domain. VR-CESM and RACMO generally agree well with ERA5 in the annual cycle of extreme precipitation over the historical time-period. One exception to this occurs in the winter months in

Baffin and Devon Islands, where the models produce lower extreme precipitation amounts than seen in ERA5 or RACMO. Conversely, they produce higher winter extreme precipitation amounts than ERA5 in SE Greenland. The models also agree well on the annual cycle of extreme precipitation when the volumetric definition (m<sup>3</sup>/day) of extreme precipitation is used (Fig. S1).

Deleted: Figure 2

Deleted: VR-CESM

Deleted: s

Formatted: Superscript

Deleted: In all months and regions and for both models.

Outside of SE Greenland, the mean extreme precipitation either remains consistent or increases in the future in all months and for both models, with increases to extreme precipitation being most acute in the warm season. Although the two models generally agree about the seasonality of changes, they disagree in SE Greenland, where VR-CESM simulations exhibit little change in any month, but RACMO simulations exhibit a marked increase in warm season extreme precipitation.

Figure 2. Mean monthly accumulation per grid cell from extreme precipitation in RACMO (blue lines), VR-CESM (orange lines), and ERA5 (black line) for the historical (1980-1998; solid lines) and future (2080-2098; dashed lines) in each subregion.

As outlined in Section 1, whether extreme precipitation falls as rain or snow has major impacts on SMB. Figure 3, shows mean monthly rain fraction of extreme and non-extreme precipitation in each model for the historical and future periods. All subregions show increases in rain fraction in the future, most of which occurs in the warm season. A sharp increase in the rain fraction in June is projected in the Canadian subregions and SW Greenland. Historically, the rain fraction was very similar between extreme and non-extreme precipitation in most subregions. This changes in the future, when several subregions show higher rain fractions on extreme precipitation days than on non-extreme days in the warm season (such as SW, CW, and NW Greenland). Historically, SE Greenland experienced a slightly lower rain fraction for extreme precipitation days than for non-extreme precipitation days in the warm season, but that difference becomes smaller in the future.

Deleted: Figure 3

Figure 3. Mean monthly rain fraction for extreme precipitation days (solid lines, "EX") and non-extreme days (dashed lines, "NON-EX") in each subregion (a-i) from RACMO (blue lines) and VR-CESM (orange lines). The darker colours show the historical averages, and the lighter colours show the future projections.

## 4 SMB Response to Extreme Precipitation

## 4.1 Mean SMB Responses

future simulations (Fig. 4; future values shown in Fig. S2). Historically, the cold season (December-March) shows positive
SMB across the domain with the highest values in SE Greenland. The two models agree well on cold season SMB, showing
minimal changes in the future simulations except for a decrease in SE Greenland. However, SE Greenland still has the highest
cold season SMB in the future projections. In the warm season historically, some low-lying and coastal regions show negative
seasonal SMB across the domain, but the negative net SMB is limited to narrow margins along the edge of ice masses. In the
future projections, the negative seasonal SMB expands to much wider margins of the GrIS, as well as the entirety of the eastern
Canadian Arctic. The models agree on overall patterns of SMB, but larger differences exist during the warm season (Fig. S3).
The higher spatial resolution of RACMO refines SMB patterns near complex topography, producing larger decreases in the
eastern Canadian Arctic and GrIS margins. RACMO also shows strong decreases in SMB reaching further inland than VRCESM. Both models showed the ablation zone similar altitudes historically (1742 m and 1830 m in RACMO and VR-CESM,

Before exploring the impact of extreme precipitation on SMB, we first consider mean seasonal SMB in the historical and

Deleted: RACMO

**Deleted:** Figure 4. Mean seasonal SMB in the region for the (a-c) cold season (DJFM) and (d-f) warm season (JJAS) for the historical period (1980-1998; a,d), future period (2080-2098; b,e), and the difference between the two periods (c,f) in RACMO.; VR-CESM shown in Fig. S1

Deleted:

Deleted: , we find little change in the mean cold season SMB

respectively), though the ablation zone in RACMO covers ~17.7% of the GrIS compared to only 9.4% in VR-CESM. However,

Figure 4. Mean seasonal SMB in the region for the (a-d) cold season (DJFM) and (e-h) warm season (JJAS) for the historical period (1980-1998; a,c,e,g), and the difference between historical and future (2080-2098) periods (FUT – HIST; b,d,f,h) in RACMO (a-b, e-f) and VR-CESM (c-d, g-h). The solid purple line denotes the top of the ablation zone for the full simulation period.

The average daily SMB on extreme and non-extreme precipitation days in the cold season in each subregion is shown in Fig. 5 to understand how extreme precipitation days differ from the average conditions. For all sub-regions, the points for every year lie above the 1:1 line, indicating that SMB is higher on extreme precipitation days than on non-extreme precipitation days. This occurs because the rain fraction is near-zero during the cold season across the domain (Fig. 3), so extreme precipitation days represent those when the most mass is added via snowfall. The largest differences between the SMB on extreme and non-extreme precipitation days are found in SW and SE Greenland which have the highest magnitude of extreme precipitation over the cold season (Fig. 2).

Deleted: c

Deleted: d

Deleted: f

Deleted: d

Deleted: d

Deleted: future period (2080-2098; b,e),

Deleted: the

Deleted: two

Deleted: c,f

Deleted: Figure 5

Deleted: Figure 3

Deleted: is

Deleted: Figure 2

Most subregions show little consistent change between HIST and FUT in the cold season. VR-CESM shows some general SMB increases on extreme precipitation days, particularly in NO Greenland. This is likely due to the increase in the magnitude of extreme precipitation events, as warmer air can hold more moisture (e.g., Bengtsson et al., 2011; Norris et al., 2019; Skific et al., 2009), which may be further enhanced by the loss of Arctic sea ice (e.g., Bintanja and Selten, 2014; Hartmuth et al., 2023; Kopec et al., 2016). This difference between HIST and FUT is not as evident in RACMO. There is further disagreement between the models in that VR-CESM produces higher SMB than RACMO in most subregions. Much of this difference may be related to the different spatial resolution of the two models. The slightly coarser resolution of VR-CESM (~ 25 km) compared to RACMO (~ 11 km) allows precipitation to penetrate further inland and affect a larger area. VR-CESM has also been shown to produce higher historical annual SMB for the GrIS compared to RACMO (van Kampenhout et al., 2020), consistent with the differences shown in Jig. 5.

Figure 5. Average DJFM Daily mean SMB on extreme days vs. non-extreme days for all subregions (a-i). Each point represents one year. RACMO is shown in blue circles and VR-CESM is represented by orange/red squares, with the darker (lighter) colour showing historical (future) means. Dashed black lines show x = 0, y = 0, and x = y.

Larger changes in SMB on both extreme and non-extreme precipitation days are projected across the domain during the warm season (Fig. 6). Historically, non-extreme precipitation days tended to have SMB near zero or weakly positive, and extreme precipitation days showed positive SMB in all subregions, with strong agreement between the two models. As in the cold

Deleted: increases in the

**Deleted:** though only small changes in the magnitude of extreme precipitation are shown in Figure 2

Deleted: {Citation}

Deleted: Figure 5

Deleted: Figure 6

season, regional SMB on warm season extreme precipitation days was greater than that of non-extreme days. Historical rain fractions remained near or below 0.25 in the warm season (Fig. 3), meaning that most extreme precipitation events resulted in mass gain via snowfall.

However, unlike the cold season, there is a large shift between the historical and future periods in the warm season. In the future projections, the SMB of both extreme and non-extreme days becomes largely negative and more variable as temperatures rise. The difference between the SMB on extreme versus non-extreme days within each season shifts in many subregions as well. Most subregions show at least some years in the future where the seasonal SMB of extreme precipitation days becomes even more negative than that of non-extreme days. The projections show that this is commonly becoming the case in regions such as SW Greenland, Baffin Island, and Ellesmere Island. Even in cases where the SMB is more positive on extreme precipitation days than non-extreme days, it is more common in the future for the SMB to be negative, with only NW and SE Greenland usually producing positive SMB on extreme precipitation days. Conversely, SW Greenland and Baffin Island shift more strongly towards extreme precipitation consistently associated with more negative SMB than its non-extreme counterparts, particularly in RACMO.

Mean SMB of extreme days [Gt w.e. / day] 0.25 0.0 -0.5 -0.25 -1.0 -1.5 -1 0 -0 5 -1.5 -1.0 -0.5 -0.25 e) CW Greenland Mean SMB of extreme days [Gt w.e. / day] 1.5 0.0 RACMO HIST -1.5 RACMO FUT -3.0 VR-CESM HIST VR-CESM FUT i) NO Greenland Mean SMB of extreme days [Gt w.e. / day] 0.75 0.00 0.0 -0.75 -1.50 -2.25 -1.50 -0.75 0.00 0.75 0.0 Mean SMB of non-extreme days [Gt w.e. / day] Mean SMB of n

Figure 6. As in Fig. 5, but for JJAS.

Deleted: However, the

Deleted: and

**Deleted:** Only SE, CW, NW, and NE Greenland continue to show extreme precipitation days remaining more positive than non-extreme days in the same year

Historically, the mean SMB of extreme and non-extreme precipitation days were relatively consistent, particularly in the warm season. In the future projections, SMB responses to warm season extreme precipitation days exhibit greater spread and variability (Fig. 6). Table 1 and Table 2 show the results of bootstrapping performed on  $IQR_{diff}$  in each subregion for the warm and cold seasons, respectively. Both RACMO and VR-CESM show a statistically significant increase in IQR in all subregions except SE Greenland in the warm season. In the cold season, VR-CESM shows an increase in IQR in NW and NE Greenland. Conversely, there is an increase in NO Greenland in RACMO, as well as a decrease in NW and SE Greenland, highlighting the disagreement between the models in the cold season. Despite these differences, both models show little overall change in the cold season SMB for extreme or non-extreme precipitation days.

Table 1. DJFM IQR bootstrapping results for each subregion. The number of events indicates the total number of extreme precipitation days in DJFM in HIST and FUT. Actual interquartile range (IQR) is the IQR of SMB anomalies on extreme precipitation days in the period and Difference indicates the difference in IQR between the two time periods. Bold indicates a statistically significant change in IQR as determined by the bootstrapping methodology outlined in Section 2.2.

|      | Subregion     | Number of  |            | Actual IQR   |       | Difference     |
|------|---------------|------------|------------|--------------|-------|----------------|
|      |               | events     |            | [Gt]         |       | (FUT-HIST)     |
|      |               | HIST       | FUT        | HIST         | FUT   | [Gt]           |
| VR-  | Baffin Island | <u>23</u>  | <u>86</u>  | 0.239        | 0.284 | 0.0 <u>45</u>  |
| CESM | Ellesmere     |            |            | <u>0.105</u> | 0.193 |                |
|      | Island        | <u>14</u>  | <u>166</u> |              |       | 0. <u>088</u>  |
|      | Devon Island  | <u>10</u>  | <u>107</u> | 0.089        | 0.056 | -0.0 <u>33</u> |
|      | SW            | <u>75</u>  | <u>135</u> | 0.923        | 0.957 | 0.034          |
|      | Greenland     |            |            |              |       |                |
|      | CW            | <u>60</u>  | <u>126</u> | 0.540        | 0.797 | 0.257          |
|      | Greenland     |            |            |              |       |                |
|      | NW            | <u>43</u>  | <u>105</u> | 0.481        | 0.748 | 0.267          |
|      | Greenland     |            |            |              |       |                |
|      | SE            | <u>150</u> | <u>137</u> | 1.905        | 2.088 | 0.183          |
|      | Greenland     |            |            |              |       |                |

Deleted: Figure 6

Deleted: Table 2 and

Deleted: Table 1

Deleted: ,

Deleted: ,

Deleted: A

Deleted: is also seen

Deleted: but it show

Deleted: s

Deleted: overall

|       | NE            | <u>131</u> | <u>163</u> | 0.982 | 1.398 | 0.415  |
|-------|---------------|------------|------------|-------|-------|--------|
|       | Greenland     |            |            |       |       |        |
|       | NO            | <u>28</u>  | 139        | 0.205 | 0.415 | 0.210  |
|       | Greenland     |            |            |       |       |        |
| RACMO | Baffin Island | 31         | 106        | 0.166 | 0.157 | -0.009 |
|       | Ellesmere     |            |            |       |       |        |
|       | Island        | 16         | 185        | 0.061 | 0.105 | 0.044  |
|       | Devon Island  | 19         | 141        | 0.011 | 0.027 | 0.016  |
|       | SW            |            |            |       |       |        |
|       | Greenland     | 59         | 66         | 0.602 | 0.976 | 0.374  |
|       | CW            |            |            |       |       |        |
|       | Greenland     | 76         | 83         | 0.355 | 0.584 | 0.229  |
|       | NW            |            |            |       |       |        |
|       | Greenland     | 49         | 119        | 0.802 | 0.366 | -0.436 |
|       | SE            |            |            |       |       |        |
|       | Greenland     | 189        | 107        | 2.072 | 1.121 | -0.951 |
|       | NE            |            |            |       |       |        |
|       | Greenland     | 127        | 150        | 0.495 | 0.607 | 0.112  |
|       | NO            |            |            |       |       |        |
|       | Greenland     | 33         | 157        | 0.157 | 0.308 | 0.151  |

Table 2. As in Table 1, but for JJAS.

| Subregion     | Number of  |            | Actual IQR |       | Difference   |
|---------------|------------|------------|------------|-------|--------------|
|               | events     |            | [Gt]       |       | (FUT-HIST)   |
|               | HIST       | FUT        | HIST       | FUT   | [Gt]         |
| Baffin Island | <u>207</u> | <u>388</u> | 0.349      | 0.455 | <u>0.106</u> |

Deleted: Table 1

| VR-   | Ellesmere     | <u>261</u> | <u>466</u> | 0.212 | 0.463 | <u>0.251</u> |
|-------|---------------|------------|------------|-------|-------|--------------|
| CESM  | Island        |            |            |       |       |              |
|       | Devon         | 233        | <u>330</u> | 0.068 | 0.174 | <u>0.106</u> |
|       | Island        |            |            |       |       |              |
|       | SW            | <u>176</u> | 311        | 1.274 | 3.521 | 2.247        |
|       | Greenland     |            |            |       |       |              |
|       | CW            | <u>163</u> | 333        | 0.590 | 1.034 | 0.444        |
|       | Greenland     |            |            |       |       |              |
|       | NW            | <u>184</u> | 418        | 0.726 | 1.137 | <u>0.411</u> |
|       | Greenland     |            |            |       |       |              |
|       | SE            | <u>76</u>  | <u>89</u>  | 1.436 | 1.852 | <u>0.416</u> |
|       | Greenland     |            |            |       |       |              |
|       | NE            | 136        | <u>273</u> | 1.010 | 1.593 | 0.584        |
|       | Greenland     |            |            |       |       |              |
|       | NO            | <u>236</u> | <u>520</u> | 0.415 | 0.880 | <u>0.466</u> |
|       | Greenland     |            |            |       |       |              |
| RACMO | Baffin Island | 194        | 428        | 0.236 | 0.516 | 0.280        |
|       | Ellesmere     |            |            |       |       |              |
|       | Island        | 226        | 531        | 0.130 | 0.409 | 0.279        |
|       | Devon         |            |            |       |       |              |
|       | Island        | 218        | 398        | 0.020 | 0.084 | 0.064        |
|       | SW            |            |            |       |       |              |
|       | Greenland     | 167        | 358        | 0.960 | 2.947 | 1.987        |
|       | CW            |            |            |       |       |              |
|       | Greenland     | 129        | 271        | 0.547 | 0.950 | 0.403        |
|       | NW            |            |            |       |       |              |
|       | Greenland     | 145        | 372        | 0.598 | 1.280 | 0.682        |

| SE        |     |     |       |       |       |
|-----------|-----|-----|-------|-------|-------|
| Greenland | 57  | 88  | 0.914 | 1.180 | 0.266 |
| NE        |     |     |       |       |       |
| Greenland | 140 | 407 | 0.623 | 1.549 | 0.926 |
| NO        |     |     |       |       |       |
|           |     |     |       |       |       |

Overall, the *IQR* changes shown in Table 1, and Table 2 confirm that the impact of extreme precipitation on SMB changes more in response to warming during the warm season than the cold season. In addition to the increased variability, it becomes more common for extreme precipitation to be associated with a negative SMB response in the future (Fig. 6). In some subregions, such as NW and CW Greenland, this means that the increased accumulation simply cannot overcome the strongly negative seasonal SMB. In other regions, such as SW Greenland and Baffin and Ellesmere Islands, this results in extreme precipitation days that are associated with more negative SMB than that of non-extreme days in the future, suggesting that the extreme precipitation days may become particularly detrimental to SMB in the future. These regions also show some of the largest increases in rain fraction (Fig. 3). This may help explain the shift towards more negative SMB associated with extreme precipitation, as rainwater directly runs-off on bare ice in ablation zones or progressively saturates firm in accumulation areas. This means that one can no longer assume that extreme precipitation directly leads to mass gain in the future climate.

#### 4.2 Seasonal Context & Change

To contextualize the importance of these events on the seasonal cumulative SMB, seasonal SMB is split into days with positive SMB  $(SMB^+_{atl})$  and negative SMB  $(SMB^-_{atl})$ , and the fraction of cumulative positive SMB  $(SMB^+_{ex\,frac})$  and negative SMB  $(SMB^-_{ex\,frac})$  that occurs on extreme precipitation days is calculated. The number of extreme precipitation days that occur with positive or negative SMB in each season are shown in Fig. S4, v.

#### 4.2.1 Cold Season

The change in  $SMB_{ex\,frac}^+$  for DJFM in RACMO is shown in Fig. 7 (future values shown in Fig. S5). Over the historical period, most of the domain received a smaller fraction of positive SMB (

Figure 7. Mean DJFM  $SMB_{ex\,frac}^+$  for HIST (1980-1998; a, c) and projected changes (FUT (2080-2098) - HIST; b, d) from RACMO (a-b) and VR-CESM (c-d),  $SMB_{ex\,frac}^-$  is zero across the domain in both periods, and is therefore not shown. The solid purple line denotes the top of the ablation zone for the full simulation period, with the future ablation zone being shown on the DIFF panels (c-d).

Deleted: ; Priestley and Catto, 2022

Deleted: from RACMO

Deleted: FUT

Deleted: . The difference (FUT - HIST) is shown in (c)

#### 4.2.2 Warm Season

More notable shifts are shown when considering changes in  $SMB_{ex\ frac}^+$  and  $SMB_{ex\ frac}^-$  in the warm season (Fig. 8, Fig. S6). Historically,  $SMB_{ex\,frac}^-$  is at or near zero across the domain, with only a small strip of coastal SW Greenland showing  $\leq 7\%$ of the negative seasonal SMB coming from extreme precipitation days. Conversely, the entire domain shows 5-20% of positive SMB during the season coming from extreme precipitation days. In the future projections, most of Greenland and northern Ellesmere Island experience an increase in  $SMB_{ex\ frac}^+$ , with extreme precipitation days contributing 10-20% more to the 455 positive SMB in the warm season than in the historical period. The opposite occurs in SW Greenland and Baffin Island, where  $SMB_{ex\,frac}^{-}$  increases at the expense of  $SMB_{ex\,frac}^{+}$ . This suggests a shift in the region, with extreme precipitation days becoming more likely to contribute to seasonal mass loss than mass gain with continued warming. This aligns with the shift towards more negative SMB associated with extreme precipitation shown in Fig. 6. This analysis cannot quantify the extent to which this shift results specifically from the precipitation itself versus other factors, such as increased temperatures on 460 extreme precipitation days. Historical case studies have estimated the direct effects of rainfall on ice to account for 

Figure 8. Mean JJAS  $SMB_{exfrac}^+$  (a-b, e-f) and  $SMB_{exfrac}^-$  (c-d, g-h) for HIST (1980-1998; a-d) and projected changes (FUT (2080-1998) - HIST; e-h) from RACMO (a, c, e, g) and VR-CESM (b, d, f, h). The solid purple line denotes the top of the ablation zone for the full simulation period, with the future ablation zone being shown on the DIFF panels (e-h).

To better understand the impacts of extreme precipitation on SMB components associated with the changes in  $SMB_{ex}^-$  and  $SMB_{ex}^+$  we explore the mean anomalies associated with warm season extreme precipitation in Fig. 9-11 (recall that the anomalies shown are calculated relative to a 31-day period centred on the extreme precipitation day, rather than the climatology, as described in Section 2.2). Mean extreme precipitation amounts and rain fraction for  $SMB_{ex}^+$  and  $SMB_{ex}^-$  from RACMO and VR-CESM are illustrated in Fig. S7 and S8, respectively. Historically, the positive SMB extreme precipitation days  $(SMB_{ex}^+)$  generally occur with positive temperature anomalies (~3-4 K) and modest anomalies in melt, runoff, and albedo (Fig. 9). While positive temperature anomalies may usually contribute to melt, the warmer air can hold more moisture and feed heavy precipitation, which is likely to fall as snow in high latitude/altitude regions during the warm season. VR-CESM

Deleted: —

Deleted: -c

Deleted: +

Deleted: d-f

Deleted: from RACMO

Deleted: , d

Deleted: b, c

Deleted: . The difference (FUT – HIST) is shown in (c) and (f) for SM and SM, respectively

Deleted: Frac

Deleted: Figures

Deleted: Figure 9

Deleted: many

(Fig. S2) shows a slight increase in refreezing occurring on positive SMB extreme precipitation days in SW Greenland. Overall, the models agree on patterns of anomalies, except for albedo, where VR-CESM shows only very small changes.

## SMB+

Figure 9. Mean anomalies on positive SMB JJAS extreme precipitation days in the historical period (1980-1998) from RACMO. Anomalies are calculated for the extreme precipitation day relative to ±15 days <u>surrounding the extreme precipitation day. The solid purple line denotes the top of the ablation zone for the full simulation period. Blue colours in each panel indicate anomalies that act to increase SMB.</u>

Next, the mean anomalies in future positive SMB extreme precipitation days  $(SMB_{ex}^+)$  are illustrated in Fig. 10 and Fig. S10, for RACMO and VR-CESM, respectively. The models agree well on the patterns of anomalies. One notable change seen in both models is that most inland regions have positive temperature anomalies historically of 2-4 K, but future projections show small negative temperature anomalies (-1 K) in some low-lying and coastal areas. This reduction in temperature anomaly associated with  $SMB_{ex}^+$  is likely due to the background increase in temperature, meaning the air can hold more moisture without requiring strong temperature anomalies. Some areas in the ablation zone show a negative future temperature anomaly associated with  $SMB_{ex}^+$  as a negative anomaly is required to bring the relatively warm summer temperatures towards freezing point, allowing for snowfall and favouring positive SMB anomalies. Both models show positive runoff anomalies of approximately 10 mm w.e. on positive SMB extreme precipitation days in SE Greenland. VR-CESM shows modest positive runoff anomalies in Ellesmere and Baffin Islands, disagreeing with the negative anomalies shown in RACMO. However, the

Deleted: 5

Deleted: Figure 10

Deleted: 6

Deleted: from

**Deleted:** the relatively warm summer temperatures mean a negative anomaly is needed

Deleted: it

Deleted:

largest differences between the models are again seen in the albedo anomalies. RACMO shows relatively large positive albedo anomalies (0.05-0.10) throughout much of the domain with decreased melt whereas VR-CESM shows very low albedo anomalies in general (anomalies below 0.025).

## SMB<sub>ex</sub>

Figure 10. As in Fig. 9, but for future (2080-2098) positive SMB JJAS extreme precipitation days from RACMO.

Some of the most notable changes exist in the negative SMB extreme precipitation days (SMB<sub>ex</sub>), which go from contributing virtually 0% of the SMB<sup>-</sup> mass loss historically to approximately 20% in the future period in coastal and southern regions of the domain in RACMO (Fig. 8g), VR-CESM also shows an increase, though of smaller magnitude (approximately 10%, Fig. 8h). The mean anomalies associated with future events are explored in Fig. 11 and Fig. S13, for RACMO and VR-CESM, respectively (historical period anomalies are shown in Fig. S11-12 as there are few occurrences, as shown in Fig. S4). While the historical simulations had limited events, one notable difference between historical and future simulations is that the temperature anomalies in the historical period (> 4 K; Fig. S7f) tended to be larger than those in the future period (< 2-3 K, and sometimes slightly negative in SE Greenland and Ellesmere Island; Fig. 11f).

The anomalies illustrated in Fig. 11 show some of the mechanisms by which extreme precipitation days result in negative SMB throughout the ablation zone. In western Greenland and Baffin Island, there are large increases in melt (Fig. 11e), which are collocated with reductions in albedo (Fig. 11d) and increased temperature (Fig. 11f). While we cannot quantify the drivers of the change in albedo, heavy rainfall may darken the surface and be a strong contributor to the negative albedo anomaly. These

Deleted: Figure 9

Deleted: Figure 11

Deleted: 9

Deleted: from

Deleted: Figures

Deleted: 7-8

Deleted: 2

regions see a modest increase in refreezing (Fig. 11c), but it does not offset the increase in melt, leading to a large increase in runoff (Fig. 11b) and negative SMB anomaly (Fig. 11a).

Both models show relatively modest SMB anomalies across most of the domain (~\_-15 mm w.e.), but larger negative anomalies in southern Greenland, occurring with large runoff increases (upwards of 30 mm w.e.). The pattern of refreezing anomalies in each model differs slightly but are relatively small (< 3 mm w.e.). Larger differences exist in albedo anomalies, where VR-CESM is near-zero across the domain and RACMO shows larger negative anomalies in SW Greenland (~-0.10) and positive anomalies along the eastern coast of Greenland (~0.05). RACMO produces much larger positive melt anomalies, which may contribute to the larger decrease in albedo, whereas VR-CESM only shows very localized increases in melt along the coast of SW Greenland. Another notable difference is that the extreme precipitation tends to reach further inland in VR-CESM than RACMO (e.g., Fig. 8g and h), likely owing to the lower resolution producing weaker topography gradients and allowing precipitation to move further inland, as found by van Kampenhout et al. (2020).

SMB<sub>ex</sub>

Figure 11. As in Fig. 9, but for future negative SMB JJAS extreme precipitation days from RACMO.

In general, the differences in positive SMB extreme precipitation day anomalies between the two time periods are modest. Conversely, the negative SMB extreme precipitation days cause notable anomalies in the future, particularly decreasing the Deleted: large

Deleted: comparing Figures

Deleted: d-f to S4d-f

Deleted: Figure 9

surface albedo in SW Greenland driving prominent increases in melt. In fact, heavy rainfall may alter snow metamorphism to darken the surface, and decreased snowfall increases the period when dark, bare ice is exposed on the surface.

#### 70 5 Discussion & Limitations

## 5.1 Connection to previous case studies

As discussed in Section 1, the effects of extreme precipitation on land ice SMB have not been investigated in a climatological context but have been explored in case studies, which can help to contextualize the results found here. Historical positive SMB extreme precipitation days are tied to increases in albedo and refreezing, with less melt occurring, similar to the effect seen by Oerlemans and Klok (2004) in the Swiss Alps. Unlike the case study presented by Oerlemans and Klok (2004), the temperature anomaly associated with warm season positive SMB extreme precipitation days in our study region remains positive during historical positive SMB extreme precipitation days, which is likely due to local climatological factors. The majority of intense precipitation events in the domain are associated with extratropical cyclones that approach from the south through Baffin Bay or along the North Atlantic Storm Track, bringing warmer air with heavy precipitation (Crawford et al., 2023; Loeb et al., 2024). Because of the high latitude, snowfall can still occur with the warmer air temperatures (Fig. 3), leading to overall mass gains. The largest positive temperature anomalies associated with extreme precipitation tend to be at higher altitudes for both

While historically, there were few negative SMB extreme precipitation days in the warm season, the future impacts align with those seen in recent case studies. Several case studies have noted large runoff anomalies associated with increased melt associated with extreme liquid precipitation in the warm season (e.g., Box et al., 2022; Doyle et al., 2015), as seen in Fig. 11. Projections suggest that refreezing will begin to decline in the future due to a lack of available firn pore space (Noël et al., 2022), which may contribute to the very modest refreezing anomalies, leading to more liquid water runoff.

#### 5.2 Model albedo differences

positive and negative SMB events.

Comparing albedo anomalies between RACMO and VR-CESM highlights large differences; RACMO produces anomalies on the order of 0.05-0.1 during extreme precipitation days, whereas those seen in VR-CESM are only ~0.01. These disparities are tied to large differences in the amount of melt that occur, suggesting that the different albedo parameterizations used may be important in understanding the responses. Both models use parts of the Snow, Ice, and Aerosol Radiative (SNICAR) model (Flanner and Zender, 2006) for snow aging metamorphism (van Dalum et al., 2022; Lawrence et al., 2018). However, other aspects of the treatment of albedo differ between the models.

One difference, for example, is the treatment of bare ice. RACMO bases the bare ice albedo on the 500 m MODerate-resolution Imaging Spectroradiometer (MODIS) albedo product, ranging between 0.30 and 0.55 (Noël et al., 2020), whereas VR-CESM

Deleted: Figure 3

Deleted: due to

Deleted: Figure 11

assumes bare ice is constant at 0.50 for the visible spectrum (van Kampenhout et al., 2020). Another notable difference is the complexity of the snow module; RACMO can represent a deep snowpack of (up to ~100 m) containing 40 layers (Noël et al., 2020) compared to the maximum depth of ~10 m made up of 12 layers in CLM5 (van Kampenhout et al., 2017, 2020).

Additionally, van Kampenhout et al. (2019) investigated the differences between native resolution CESM and VR-CESM in reproducing historical GrIS SMB and noted several potential biases related to albedo representation. One such issue is that CLM5 repartitions precipitation phase from CAM based on temperature, which does not allow for supercooled rainfall that darkens surface albedo, particularly for the northern GrIS. The downscaling also redistributes clouds within the simulation, which was found to delay summer melt. Additionally, CLM5 does not account for changes in snow properties due to pooling water on the surface, which can lead to darkening being missed by the model. Each of these factors can lead to higher albedos and reduced melt in CLM5, reducing the melt-albedo feedback. This would lead to smaller albedo changes, as seen in Fig. 89-10 and S12-13.

Deleted: Figures

Deleted: 5-6 Deleted: 7-8

Further differences in albedo may arise from the difference in the irreducible water saturation thresholds between the models. While the difference is relatively minor (2% versus 3.3% in RACMO and VR-CESM, respectively), a higher threshold can result in slightly lower runoff occurrence. Even a modest change in simulated runoff can have a variety of impacts, since liquid water at the surface can alter snow metamorphism, albedo, and melt. Glaude et al. (2024) hypothesized this to be a factor in major differences in GrIS SMB projections found from three commonly used regional climate models, including RACMO.

5.3 Limitations 620

> The results presented here help to illustrate the impacts and importance of extreme precipitation events on seasonal SMB, but there are several notable limitations. Firstly, across the domain, it is common for extreme precipitation to occur with warm air advection, driven by features such as atmospheric rivers (e.g., Box et al., 2022; Loeb et al., 2024). Increased air temperature alone can cause increased melt and drive some of the anomalies seen in Section 4. Because of this, it is difficult to disentangle the effects of other climate variables from the effects of extreme precipitation. Indeed, the changes illustrated here are likely small contributors to the total decrease in SMB from melt due to rising temperatures but can still provide a better understanding of processes impacting the SMB. Future work analysing the surface energy balance would allow for a more detailed understanding of the magnitude of the impacts associated with the precipitation itself versus other factors on extreme precipitation days.

Deleted: deeper

Additionally, this analysis only considers impacts on the day of each extreme precipitation event, but the impacts may extend beyond. For example, extreme precipitation events can have direct effects on SMB that last for several days, such as albedo changes (e.g., Bailey and Hubbard, 2025; Oerlemans and Klok, 2004), which may lead to differing seasonal-scale impacts. The calculation of anomalies relative to ± 15 days centred on the extreme precipitation day means that these multi-day impacts are included in the background mean, suggesting that the anomalies shown in Fig. 9-11 may be slightly underestimated. This

likely has the largest impact on the albedo anomalies. We also only consider impacts within the area experiencing extreme precipitation, but it is also possible for the precipitation to affect SMB beyond the precipitation area. For example, increased runoff from rainfall and melt can lead to increased melt or refreezing downslope, which would not be accounted for in the current analysis. Future work investigating these extended impacts is necessary to better quantify the true importance of extreme precipitation events.

Finally, only two simulations with relatively short time periods are analyzed in this study, although agreement between the two separate models helps increase confidence in the conclusions. Glaude et al. (2024) illustrated large differences in annual GrIS SMB from three commonly used polar regional climate models using the same forcing data, including the RACMO simulation used in this study. Even though the same CESM2 forcing dataset is used, the three regional models yielded annual SMB that differed by a factor of two, highlighting the importance of looking at a range of projections to understand potential outcomes. RACMO produced the highest future SMB of the three simulations, suggesting that the impacts seen in this study may be more intense in simulations from different polar climate models. Repetition of this assessment with a larger ensemble of high-resolution models with longer simulation periods would be valuable to further substantiate results. It would be particularly insightful to explore models with differing albedo parameterizations to further explore the albedo-related differences seen between RACMO and VR-CESM. Additionally, using higher spatial resolution models may better resolve extreme precipitation events (Ali and Tandon, 2024; Cai et al., 2018) and SMB processes (e.g., Noël et al., 2016).

## 6 Conclusions

Through the presented analysis of the impacts and importance of extreme precipitation days on the SMB of land ice in Greenland and the Eastern Canadian Arctic, we come to three main conclusions:

Firstly, the changes that occur during the warm season (JJAS) are more prominent than those of the cold season (DJFM), having the potential for larger implications for seasonal SMB. Historically, precipitation days in the warm season had positive average SMB in virtually all years and subregions except for SW Greenland and Baffin Island. However, as the climate warms, much of the domain shifts to almost all precipitation days being associated with negative SMB. Even extreme precipitation days are projected to always result in a mean negative seasonal SMB in SW Greenland and the Canadian subregions in the future. There is also a shift in the role that extreme precipitation days play, in these subregions in the future. In the historical period, the mean SMB of extreme days was always higher (more positive) than on non-extreme precipitation days. The future projections indicate that this may no longer be the case in SW Greenland and Baffin Jsland, where mean SMB on extreme days becomes even more negative than non-extreme days, particularly in RACMO. This likely results from the shift towards rainfall at the expense of snowfall as temperatures rise. In addition to the potential surface darkening, heavy rainfall can lead to

**Deleted:** events

Deleted: s

Deleted: and Ellesmere

Deleted: s

dramatic runoff increases and pooling water that drives further melt. Overall, model projections suggest that extreme precipitation days shift from being <u>consistent</u> contributors of warm season mass gain to a potential driver of sustained mass loss in the future in regions such as SW Greenland.

Secondly, the relative importance of extreme precipitation days to seasonal <u>positive and negative</u> SMB <u>components</u> is projected to increase in the <u>across much of the domain.</u> The warm season illustrates both positive and negative changes across the domain; extreme precipitation days account for a larger portion of warm season  $SMB^+$  across inland regions and  $SMB^-$  in coastal regions in the <u>ablation zone</u>, particularly in SW Greenland where the contribution of extreme precipitation days to negative SMB increases from near-zero to approximately 20%. Future changes are generally smaller in the cold season, when the most notable change is a decrease in the contribution of extreme precipitation days to positive SMB in SE Greenland.

Small increases across the northernmost regions of the domain reflect the increased water vanour holding capacity of warmer.

Small increases across the northernmost regions of the domain reflect the increased water vapour holding capacity of warmer air, which allows for more cold season extreme precipitation, and may also be facilitated by sea ice loss and enhanced moisture availability. These changes result in most of the domain showing ~5-10% of cold season SMB coming from extreme precipitation days in the future

Finally, the SMB responses to warm season extreme precipitation are projected to become more variable in the future. Both models show increases in the IQR of SMB anomalies on extreme precipitation days everywhere except for SE Greenland, where cold season changes are more prominent. The warm season shows the largest projected shift in rainfall fraction. This can drive the more varied SMB impacts in the future since the effects of an extreme event can be dramatically different depending on the precipitation phase. Combined with the shift towards negative SMB, this suggests that one can no longer assume that extreme precipitation simply leads to a mass gain in the region.

This work provides a first estimate of the seasonal-scale impacts of extreme precipitation on the SMB of glaciers and ice caps in the eastern Canadian Arctic and Greenland and how that role may change in the future. While only two models are used in this analysis, it provides a framework for future studies using larger ensembles to further investigate the contribution of extreme precipitation to land ice SMB anomalies under climate warming.

#### **Data Availability Statement**

680

685

The CESM2-forced RACMO historical reconstruction and future projection under SSP5-8.5 are discussed in Noël et al. (2020), and can be freely accessed from Brice Noël (bnoel@uliege.be) upon request and without conditions. Processed VR-CESM data is available on the Canadian Watershed Information Network (CanWIN, DOI forthcoming).

**Deleted:** warm season, with smaller changes occurring during the

Deleted:

Deleted: with

Deleted: s

Deleted: where

Deleted: as

Deleted: the loss of

Deleted: may promote more intense precipitatio

Deleted: n

Formatted: Normal

#### 715 Author contributions

NAL, AC, and JS developed the study and methodology. BN shared RACMO data and guidance on analysis. NAL performed analysis and prepared manuscript. All authors contributed to editing the manuscript.

## Acknowledgments

This research was undertaken, in part, thanks to funding from the Canada 150 Research Chairs Program (Grant 50296), the
Natural Sciences and Engineering Research Council of Canada (NSERC), and Horizon 2020 CRiceS (Grant 101003826). B.
Noël is a Research Associate of the Fonds de la Recherche Scientifique de Belgique – F.R.S.-FNRS. The authors would like
to thank Jan Lenaerts and the Land Ice Working Group at the National Center for Atmospheric Research for VR-CESM run
support, and two anonymous reviewers for their constructive feedback.

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
