# Peer review of "Modelling the Impacts of Historical and Future Extreme Precipitation Days on Seasonal Surface Mass Balance in the Eastern Canadian"

_EGUsphere, 2025_

## Author Comment (AC2)

**Response to Reviewers**

We would like to thank the two anonymous reviewers for their helpful feedback in improving this manuscript. Below is a summary of some of the major updates to the manuscript:

- Based on comments from both reviewers, the title has been updated to "Modelling the Impacts of Historical and Future Extreme Precipitation Days on Seasonal Surface Mass Balance in the Eastern Canadian Arctic and Greenland" to better reflect the material in the manuscript.
- Figures 4, 7, and 8 have been edited to add VR-CESM results, so only the historical and change (FUT-HIST) panels are now shown. The future panels for both models are now available in the supplementary information.
- During the revisions, it was realized that land fraction wasn't used for VR-CESM area integrated calculations – this has been corrected, which reduced some values slightly to account for the cells not entirely over land. This resulted in very slight shifts to values in Tables 1 and 2, and Figures 5 and 6, but there isn't meaningful change to any interpretations other than that the change in IQR in NO Greenland in the cold season is no longer statistically significant.
- A minor calculation error found in calculation of mean monthly extreme precipitation amounts in RACMO (Figure 2) – now updated, showing closer agreement between VR-CESM and RACMO

Additionally, some wording was made more precise regarding methods. Specific reviewer comments are addressed below.

Note: reviewer comments = bold, author responses = normal text, added text to manuscript = italics

**Anonymous Reviewer 1**

**This manuscript investigates the impacts of extreme precipitation events on seasonal historical and future SMB in the Eastern Canadian Arctic and Greenland using simulations from the RACMO and VR-CESM models. Comparing extreme precipitation days with non-extreme precipitation days, one of the main findings is that historically extreme precipitation days consistently leads to a higher SMB, both in the cold and warm seasons, while for the future this relationship only persists for the cold season. For the warm season, the extreme precipitation results in a less positive/more variable SMB and its contribution becomes more prominent. Further because of a shift towards more rainfall, extreme precipitation days increasingly coincide with mass loss, particularly in SW Greenland and Baffin Island.**

**The manuscript is well structured and pleasant to read. The content of the manuscript is very interesting and fits well in the scope of the journal. However, the manuscript needs some improvements here and there. I have added several comments/suggestions that may help the authors to improve their manuscript.**

**Title:**

- **Although the title itself is good, it can use a bit more improvement to reflect the content of the manuscript better. Personally, I was thinking about: "Modelling the Impacts of Extreme Precipitation Events on Seasonal Historical and Future Surface Mass Balance in the Eastern Canadian Arctic and Greenland"**
  Great suggestion – we have edited the title to "Modelling the Impacts of Historical and Future

Extreme Precipitation Days on Seasonal Surface Mass Balance in the Eastern Canadian Arctic and Greenland" to more accurately reflect the material in the paper.

**Section 1: Introduction:**

- **L75-80: The authors mention an example of how an extreme snowfall event in the Swiss Alps can impact albedo and surface melt for several days, but can they also give an example of heavy snowfall events and related impacts in Greenland and or the Eastern Canadian Arctic? In my opinion, these kinds of examples contribute better to the (geographical) settings described in this manuscript.**
  The Oerlemans & Klok Swiss Alps example has been changed to a recent paper published analyzing an atmospheric river event in SE Greenland that resulted in notable mass gain in the region.

  Lines 87-93: "*Bailey & Hubbard (2025) presented an analysis of a March 2022 atmospheric river event impacting the southeastern GrIS. Temperatures remained near 0 ℃, resulting in heavy snowfall across the region. The fresh snowfall increased surface albedo, delaying the onset of the melt season by 11 days. The effect of the reduced melt and added snow mass from the event was found to offset seasonal mass loss by approximately 8% during the following melt season.* While extreme precipitation events can cause dramatic short-term SMB changes, their importance in a seasonal context has not been studied. While extreme precipitation events can cause dramatic short-term SMB changes*, either positive or negative*, their importance in a seasonal context has not been *explicitly* studied."

**Section 2: Data and Methodology:**

- **L123-124: The sentence "The snow cover … (Lawrence et al., 2019)" is redundant as the authors already explain the snow model in CLM5 in L133-134. Therefore, the sentence here can be removed.**
  We have removed this sentence.

- **L124-L126: This sentence needs to be rephrased a bit as elevation downscaling is only applied over glacierized land units and not over the entire grid cell. Also, it is relevant to mention (in my opinion) that CLM uses a subdivision scheme that heterogeneously subdivides grid cells into several land units to account of the heterogeneity of the land surface. This kind of information is maybe less relevant for the GrIS itself but could be relevant for interpreting results in the ablation zones and/or partially glaciated grid cells.**

  The sentence has been edited slightly to specify that it applies only to glaciated grid cells and have also added a new sentence to describe the sub-grid cell accounting for heterogeneity in the land surface (line 145-149): "To account for the complex topography in glaciated areas, each *glaciated* grid cell is divided into 10 elevation classes to adjust atmospheric surface temperature, potential temperature, specific humidity, density, and pressure over ice surfaces (Lawrence et al., 2019). *Along the periphery of ice caps and the GrIS, grid cells are also sub-divided into different land types to account for surface heterogeneity.*"

- **L133: Please add 10 m water equivalent (w.e.) or 10 m w.e.**
  Added – "The snow model within CLM5 contains up to 12 layers, representing up to 10 m *water*

*equivalent (w.e.)* of firn or snow (van Kampenhout et al., 2017; Lawrence et al., 2019)."

- **L155: What is the main reason for defining extreme precipitation in 2 different ways? Please explain in the manuscript.**
  This was done to allow for analysis at both the grid cell- and subregional-level – using the grid cell-based definition is ideal for highlighting the spatial gradients in any given result, but it sometimes fails to highlight widespread events. For example, the extreme thresholds may vary substantially throughout the region depending on topography and climatic factors. The regional (volume-based) threshold allows us to consider such events in the context of an entire GrIS drainage basin.

  The relevant text (lines 184-186) now reads: *"by individual grid cell, which highlights spatial gradients in extreme precipitation and its impacts, and by subregion, which facilitates the analysis of events that are extreme over an entire drainage basin or island."*

- **L157-158: Could the authors elaborate more on the meaning of "total daily precipitation volume over all grid cells"? Is it the areal mean of daily precipitation expressed in mm/day (i.e. averaged over all grid cells)?**
  Here we are referring to the total volume of precipitation integrated over all ice cells in the subregion (resulting in a total volume of extreme precipitation in $m^3$) – an explanation clarifying this has been added (lines 187-190): *"At the subregion level, extreme precipitation days are defined as the days at or above the 95th percentile of total daily precipitation volume ($m^3$/day) in the subregion, calculated by summing the product of precipitation and glaciated grid cell area for all grid cells in a subregion."*

- **L160: I assume that the extreme threshold would be the 95$^{th}$ percentile of total daily precipitation at subregional level?**
  Yes, and this should be more apparent with the edit made, as described in the response to the previous comment: *"...calculated by summing the product of precipitation multiplied by grid cell area for all glaciated grid cells in a subregion."*

- **L166: Is the window of +/- 15 days centered around each extreme precipitation day? If so, please indicate that in the manuscript. And why do the authors choose for a window of +/- 15 days. Does the calculation of anomalies in this way not increase the risk of including values (for SMB and its components) for non-extreme days and therefore cause a mixed/noisy signal?**
  We're only using +/-15 days to calculate the baseline reference, so the anomalies presented are the extreme day minus the average of +/- 15 days surrounding the extreme day. This 31-day baseline period was selected as it is long enough to provide an approximation of average conditions at the time an event occurs to illustrate how impactful that day is. The text has been modified as follows to clarify the method used:

  Lines 197-201: "*Short term anomalies in SMB-related variables for each extreme precipitation day were calculated relative to a window of ±15 days centred on the extreme precipitation day. We selected a 31-day period as the baseline instead of a climatology to focus on the within-season anomaly. This removes effects of background changes in mean seasonal/annual SMB conditions but will underestimate anomalies when the extreme events' impacts on SMB variables last for several days, which is likely most common for the albedo anomalies (e.g., Bailey and Hubbard, 2025;*

*Oerlemans and Klok, 2004)."*

- **L171-173: This sentence is not clear. So, if I understand well the future IQR increases are statistically significant if the real IQRdiff is greater than the IQRdiff in 975 of the 1000 tests (or maybe in other words the 97.5ᵗʰ percentile of all IQRdiff values)? And why 975 tests are chosen as the threshold?**
  That is correct – if the real IQRdiff is either greater or less than 975/1000 tests, there is a statistically significant change in the IQR. The value of 975 was selected because it yields a 95% confidence level in the statistic (i.e., a two-tailed 95% confidence interval, as it is only statistically significant if the real IQRdiff is in the lowest or highest 2.5% of IQRdiff tests).

  Some additional text clarifying this has been added to lines 219-222: *"Repetitions were performed 1000 times, and if the real $IQR_{diff}$ was greater than (respectively less than) 975 of the tests, this indicated a statistically significant increase (respectively decrease) in IQR in the future simulation, yielding a two-tailed confidence interval of 95%."*

**Section 3: Extreme Precipitation:**

- **L191-193: I guess also SE Greenland forms an exception in the winter months but then in an opposite way with RACMO/ERA5 extreme precipitation amounts being lower than VR-CESM extreme precipitation amounts.**
  Great observation – Upon fixing the calculation error with RACMO, it actually agrees with VR-CESM here, but they both disagree with ERA5, so a sentence with this point has been added on lines 239-241: *"One exception to this occurs in the winter months in Baffin and Devon Islands, where the models produce lower extreme precipitation amounts than seen in ERA5 or RACMO. Conversely, they produce higher winter extreme precipitation amounts than ERA5 in SE Greenland."*

- **Figure 2 shows the mean monthly accumulation per grid cell for the defined domains. Does the mean monthly accumulation based on extreme precipitation at subregional-level as defined in L157-158 look different?**
  Good question! We decided to use the grid cell-based definition here because it is based on millimeters of precipitation accumulation, whereas the regional definition is a water volume integrated over the entire region. Since it is easier to understand the magnitude of extreme precipitation accumulations in terms of mm w.e., it seemed more appropriate and meaningful here to use this definition. The same figure showing the mean monthly accumulations based on the regional volume-based definition is shown below.

[Figure]

This figure has been added to the supplement (Figure S1) since using this definition also results in good agreement between RACMO and VR-CESM in most regions. Text has been added to lines 241-243: "Conversely, they produce higher winter extreme precipitation amounts than ERA5 in SE Greenland. *The models also agree well on the annual cycle of extreme precipitation when the volumetric definition of extreme precipitation is used (Figure S1).*"

**Section 4: SMB Response to Extreme Precipitation:**

- **L218: "…we first consider…" instead of "…we consider…" (I assume the authors will consider mean seasonal SMB from VR-CESM simulations in a later stage).**
  Added

- **Figure 4; L218-226: To what extent differs the simulated/projected SMB in VR-CESM from those simulated/projected in RACMO? Please explain in the manuscript. Further I suggest the authors to merge Figure 4 and Figure S1 into one Figure. To do this I recommend removing the FUT fields as the FUT-HIST fields already show the differences in the future relative to the historical period. Once the FUT fields are removed it is possible to make one figure with 2x4 panels where the rightmost panels show the results for VR-CESM, and the leftmost panels show the results for RACMO (or other way around). In my opinion showing the VR-CESM SMB fields in Figure 4 is also important for understanding the results shown in the figures that follow.**
  Below is an updated version of Figure 4 showing the seasonal SMB for the cold season (a-d) and warm season (e-h) from RACMO (a-b, e-f) and VR-CESM (c-d, g-h). The DIFF panels (b, d, f, h) represent FUT – HIST. A line has been added to each panel showing the ablation zone for the respective simulation. Figure 4 has been updated to this new version in the manuscript.

[Figure]

*Figure 4. Mean seasonal SMB in the region for the (a-d) cold season (DJFM) and (e-h) warm season (JJAS) for the historical period (1980-1998; a,c,e,g), future period (2080-2098; b,d,f,), and the difference between historical and future (2080-2098) periods (FUT – HIST; b,d,f,h) in RACMO (a-b, e-f) and VR-CESM (c-d, g-h). The solid purple line denotes the top of the ablation zone for the full simulation period.*

A supplementary figure has also been added showing the differences between the mean SMB for each period between the two models and is shown below. Here, the green (purple) colours indicate that VR-CESM (RACMO) has a higher seasonal mean SMB. This highlights that the models agree quite well in the cold season, but larger differences exist in the warm season (particularly in the future).

The differences in the historical warm season (panel c) look like they can largely be connected to the improved spatial resolution of RACMO – differences are largely concentrated to regions of complex topography. RACMO shows stronger decreases in SMB across the northernmost areas of the domain, the Canadian subregions, and up to higher altitudes in the ice sheet. Historically, the ablation zone in RACMO covers 17.7% of the GrIS, while in VR-CESM it only covers 9.4% of the area of the ice sheet, with maximum ablation zone altitudes reaching 1830 m and 1742 m in VR-CESM and RACMO, respectively. In the future period, both models show the ablation area expand to cover an additional 28% of the ice sheet – RACMO shows the ablation zone reaching altitudes up to 2658 m, compared to a maximum altitude of 2297 m in VR-CESM.

[Figure]

The following text has been added to Section 4.1 to highlight the similarities and differences between the two models (Lines 280-296): "*The models agree on overall patterns of SMB, but larger differences exist during the warm season (Fig. S3). The higher spatial resolution of RACMO refines SMB patterns near complex topography, producing larger decreases in the eastern Canadian Arctic and GrIS margins. RACMO also shows strong decreases in SMB reaching further inland than VR-CESM. Both models showed the ablation zone similar altitudes historically (1742 m and 1830 m in RACMO and VR-CESM, respectively), though the ablation zone in RACMO covers ~17.7% of the GrIS compared to only 9.4% in VR-CESM. However, in the future, both models show the ablation zone expanding to cover an additional ~28% of GrIS area. RACMO shows expansion of the ablation zone to altitudes up to 2658 m, compared to only 2297 m in VR-CESM. These differences between RACMO and VR-CESM are consistent with those found by van Kampenhout et al. (2019), which also showed the largest differences in the ablation zone with VR-CESM producing higher SMB than RACMO.*"

- **L239-241: Both RACMO and VR-CESM experience an increase in temperature. Then why VR-CESM shows a more pronounced increase in extreme precipitation days than RACMO? Is it because VR-CESM experiences a stronger increase in temperature which leads to stronger increases in atmospheric moisture content? Or is the difference explained by other phenomena related to the atmosphere or maybe even sea ice? For example, what happens with the zonal wind patterns (for example at 850 hPa) during the cold season? Could it be possible that future zonal wind patterns (i.e. related to the jet stream) display a stronger poleward displacement in VR-CESM than in RACMO and therefore explain the SMB increases in NO Greenland?**
A calculation error was found in the values for RACMO shown in Figure 2 (mean monthly extreme

precipitation accumulations) – upon fixing the error, the values in RACMO and VR-CESM are much closer in most cases (updated figure shown below). In fact, RACMO often is slightly higher than VR-CESM, but the values do not differ drastically.

[Figure]

- **L262: Could the authors elaborate more on what they mean with: "the difference between the SMB on extreme and non-extreme days shifts in many subregions as well"? Are the authors trying to say that the trendlines for future and historical simulations have the same slope but another intercept (i.e. lower intercept)?**
  Here, we are referring to the fact that most subregions begin showing some years where the seasonal SMB on extreme precipitation days is more negative than that of non-extreme days of the same season (i.e., dots on the figure falling below the 1-to-1 line).

  The text has been revised to try to clarify this point on lines 349-354: "In the future projections, the SMB of both extreme and non-extreme days becomes largely negative and more variable as temperatures rise. *The difference between the SMB on extreme versus non-extreme days within each season shifts in many subregions as well. Most subregions show at least some years in the future where the seasonal SMB of extreme precipitation days becomes even more negative than that of non-extreme days. The projections show that this is commonly becoming the case in regions such as SW Greenland, Baffin Island, and Ellesmere Island.*"

- **Figure 5 and 6: In my opinion it could have an added value to insert trendlines in the figure so readers can more easily derive the ratio/relation between extreme day SMB and non-extreme day SMB for both models and periods.**
  Below are Figures 5 and 6 for DJFM and JJAS, respectively, with trendlines added. This is an interesting idea, however the inclusion of trendlines does make the figures busier and may make interpretation more difficult due to clutter. Specifically, having the regression lines makes it more difficult to discern where the points are relative to the 0-lines and whether they are above or below the 1:1 line, which are the aspects we focus on in the text.

[Figure]

- **262-264: I would suggest rephrasing the sentence to: "Only SE, CW, NW, and NE Greenland continue to show a more positive SMB on extreme precipitation days than the SMB on non-extreme precipitation days (in the same year)". I think the latter part is a little redundant.**
  This sentence has been revised to instead focus on the changes rather than which subregions show less change (lines 349-354): "In the future projections, the SMB of both extreme and non-extreme days becomes largely negative and more variable as temperatures rise. *The difference between the SMB on extreme versus non-extreme days within each season shifts in many subregions as well.*

*Most subregions show at least some years in the future where the seasonal SMB of extreme precipitation days becomes even more negative than that of non-extreme days. The projections show that this is commonly becoming the case in regions such as SW Greenland, Baffin Island, and Ellesmere Island*."

- **L274, L287: "Table 1 and 2…" instead of "Table 2 and Table 1…"**
  Corrected

- **L275, Table 1 and 2: What is statistically significant according to the authors? Is that statistically significant at the 95% confidence interval? And how is the statistical significance of changes/differences defined/calculated? Via a student's t-test?**
  The statistical significance is defined by the bootstrapping methodology described in Section 2.2. The caption of Table 1 has been edited to include this information: *"Table 1. DJFM IQR bootstrapping results for each subregion. The number of events indicates the total number of extreme precipitation days in DJFM in HIST and FUT. Actual interquartile range (IQR) is the IQR of SMB anomalies on extreme precipitation days in the period and Difference indicates the difference in IQR between the two time periods. Bold indicates a statistically significant change in IQR as determined by the bootstrapping methodology outlined in Section 2.2."*

- **L288: Please remove the text "Figure 6".**
  Corrected

- **L298-299: Are the days with positive SMB (SMB+) referring to all days with positive SMB (SMB+ all) or to extreme precipitation days with positive SMB (SMB+ ex)? To avoid confusion, I recommend the authors to use the abbreviations as defined in Section 2.2.**
  Here we were trying to provide a brief reminder of how we split the seasonal SMB into the positive and negative components based on whether the daily SMB is positive or negative (so $SMB_{all}^+$ and $SMB_{all}^-$), since there is a lot of analysis presented between the methods description and Section 4.2.

  The abbreviations have been edited to match those defined in Section 2.2 (lines 407-409): *"To contextualize the importance of these events on the seasonal cumulative SMB, seasonal SMB is split into days with positive SMB ($SMB_{all}^+$) and negative SMB ($SMB_{all}^-$), and the fraction of cumulative positive SMB ($SMB_{ex\ frac}^+$) and negative SMB ($SMB_{ex\ frac}^-$) that occurs on extreme precipitation days is calculated.*"

- **Figures 7 and 8: Similar to Figure 4 I strongly suggest removing the FUT fields and instead adding the VR-CESM fields for HIST and FUT-HIST as these also include the key findings that are described in the manuscript.**
  Figures 7 and 8 have been edited to include both RACMO and VR-CESM, with the future values being shown in the supplement. The updated figures are shown below.
  Figure 7:

[Figure]

Figure 8:

[Figure]

- **L311-313: Could the reduction in SE Greenland be somehow related to the increased SMB on extreme precipitation days in NO Greenland? As mentioned in one of my earlier comments I can imagine that a poleward displacement of zonal wind patterns could be responsible for these changes, that is a decrease in extra-tropical cyclones/moisture inflow in SE Greenland, but an increase in moisture inflow in NO Greenland.**

Interesting question – it is possible that there is a connection between the cold season reduction in extreme precipitation day SMB in SE Greenland and the increase in NO Greenland but we cannot say for sure whether or not this is the case. We did analyze cyclone frequency in the VR-CESM simulations (results presented in Loeb et al. (2024)), and the changes would not conclusively suggest this poleward shift. There is a slight reduction in cyclone frequency near SE Greenland and increase in northern Baffin Bay, but it is likely that much of the changes are tied to increased precipitation amounts falling from a similar number of storms (as suggested by Yettella and Kay, 2017). Additionally, Huai et al. (2025) found that the decrease in SE Greenland precipitation is likely tied to the northeastward shift of the Icelandic Low, whereas northern precipitation increases are largely tied to increased moisture availability as sea ice declines.

References:

Huai, B., Ding, M., van den Broeke, M. R., Reijmer, C. H., Noël, B., Sun, W., & Wang, Y. (2025). Future large-scale atmospheric circulation changes and Greenland precipitation. Npj Climate and Atmospheric Science, 8(1), 10. https://doi.org/10.1038/s41612-025-00899-z

Loeb, N. A., Crawford, A., Herrington, A., McCrystall, M., Stroeve, J., and Hanesiak, J.: Projections and Physical Drivers of Extreme Precipitation in Greenland & Baffin Bay, Journal of Geophysical Research: Atmospheres, 129, e2024JD041375, https://doi.org/10.1029/2024JD041375, 2024.

Yettella, V. and Kay, J. E.: How will precipitation change in extratropical cyclones as the planet warms? Insights from a large initial condition climate model ensemble, Clim Dyn, 49, 1765–1781, https://doi.org/10.1007/s00382-016-3410-2, 2017.

- **329: Please remove the text "Figure 8".**
  Corrected

- **L335-336: Please remove the text "Figure 9". Also, could the authors add (SMB+ ex) between brackets after "extreme precipitation days" for clarification? I presume it is SMB+ ex and not SMB+ all.**
  "Figure 9" has been removed and "$(SMB_{ex}^{+})$" has been added

- **L336: Although I can understand the relation between positive SMB and positive temperature anomalies in the Greenland area, readers could question about why positive temperature anomalies coincide with a positive SMB as they could expect the SMB to be negative (positive temperature anomalies → more melt → negative SMB). Therefore, it could be useful to explain in more detail how the authors interpret the coincidence of positive SMB and positive temperature anomalies (or refer to Section 5.1 where the authors explain these findings in more detail).**
  Good point – a sentence has been added to address the connection between positive temperature anomalies with positive SMB related to increased precipitation so readers don't have to wait until Section 5.2.

  Lines 505-509: "*This reduction in temperature anomaly associated with $SMB_{ex}^{+}$ is likely due to the background increase in temperature, meaning the air can hold more moisture without requiring strong temperature anomalies. Some areas in the ablation zone show a negative future temperature anomaly associated with $SMB_{ex}^{+}$ as a negative anomaly is required to bring relatively warm summer air temperatures towards the freezing point, allowing for snowfall and favouring positive SMB anomalies.*"

- **Figures 9-11 and S5-S9: Please change SMB+ (or SMB-) to SMB+ ex (SMB- ex). Also, I presume the window of +/-15 days used for the calculation of the anomalies is centered around the extreme precipitation day?**
  Corrected, and the following text has been added to Figure 9's caption to clarify the anomaly calculation period: "Mean anomalies on positive SMB JJAS extreme precipitation days in the historical period (1980-1998) from RACMO. Anomalies are calculated for the extreme precipitation

day relative to ±15 days *surrounding that extreme precipitation day.*"

- **L343 and L356: "for RACMO and VR-CESM…" instead of "from RACMO and VR-CESM…".**
  Corrected

- **L355: SMB- ex instead of SMB-?**
  Corrected

- **L356: I presume Fig. 8 is referring to Fig. 8f?**
  Since Fig. 8 has been updated to include VR-CESM, the sentence has been edited to refer to 8g (RACMO DIFF) and an additional statement has been added to highlight the difference in VR-CESM (lines 525-528): "Some of the most notable changes exist in the negative SMB extreme precipitation days ($SMB_{ex}^-$), which go from contributing virtually 0% of the $SMB^-$ mass loss historically to approximately 20% in the future period in coastal and southern regions of the domain *in RACMO (Fig. 8g). VR-CESM also shows an increase, though of smaller magnitude (approximately 10%, Fig. 8h).*

- **L363: "…large runoff increases…" instead of "…runoff large increases…".**
  Corrected

- **Figures 9-11 and S5-S9: In my opinion it could also have an added value to add the rainfall and snowfall fields associated with positive and negative SMB extreme precipitation days. That also supports the findings described in L373-376.**
  Below are Figures 9-11 updated to include total precipitation and snowfall anomalies in panels b and c, respectively.

Figure 9 (hist, SMB+):

[Figure]

Figure 10 (future, SMB+):

[Figure]

Figure 11 (future, SMB-):

[Figure]

The precipitation anomalies are somewhat difficult to interpret, and mean values on extreme precipitation days are likely a lot more meaningful to illustrate what kind of precipitation is occurring on positive versus negative SMB extreme precipitation days. Because of that, we've opted to instead add supplementary figures showing the mean extreme precipitation amounts and mean rain fractions on $SMB_{ex}^{-}$ and $SMB_{ex}^{+}$ days in the warm season, shown below.

RACMO:

[Figure]

VR-CESM:

[Figure]

Section 5: Discussion & Limitations:

- **L430-432: I don't entirely get the point of the authors as the authors already use +/- 15-day anomalies for SMB and its components for an extreme precipitation day. In that way I can imagine that the effects of extreme precipitation beyond the day itself are already included?** Using the +/- 15-day period surrounding the extreme precipitation day to calculate the anomalies means that the days following the extreme day are included in the background/mean conditions that the extreme day is compared to. Since previous studies have indicated potential for relative long-lasting results (multiple days (e.g., Oerlemans & Klok, 2004) to weeks/months (e.g., Bailey & Hubbard, 2025)), this means that some of the impacts beyond the day of the event would be included in the background mean in the anomaly calculation. Assuming that the lasting effects are in the same direction as on the extreme day, this may potentially reduce the calculated anomaly on the day of the event (in addition to not being explicitly considered in this analysis).

**Data availability:**

- **I miss a data availability statement in this manuscript. Could the authors include one?**
  The data availability statement has been added: "The CESM2-forced RACMO historical reconstruction and future projection under SSP5-8.5 are discussed in Noël et al. (2020), and can be freely accessed from Brice Noël (bnoel@uliege.be) upon request and without conditions. Processed VR-CESM data is available on the Canadian Watershed Information Network (CanWIN, DOI forthcoming)."

**Supplementary Data:**

- **Could the authors prevent overlap between text and panels as shown in Figures S5-S6 and S8-S9?**
  Corrected

**Anonymous Reviewer 2**

**Summary**

**In this study, the authors analyze the historical and future contributions of extreme precipitation events to the surface mass balance (SMB) of ice masses in Greenland and the eastern Canadian Arctic Archipelago. They use two distinct sets of SSP5-8.5 simulations (CESM downscaled with RACMO and its multi-layer snow module; CESM-VR with the CLM5 land model). They find that in the historical period, extreme precipitation days are associated with increases in SMB during both the cold and warm season, but in future warming scenarios, mass loss becomes more common on extreme precipitation days due to an increase in the proportion of liquid precipitation, particularly in some favored areas such as southwest Greenland and Baffin Island.**

**Overall, the paper is well written with clear explanations and figures. I have several comments that are primarily requests for the authors to better explain and contextualize their findings, especially with regard to the physical processes that cause melt on extreme precipitation days. Provided these comments are addressed, overall I think this study will provide an important and novel contribution to projections of the future evolution of the Arctic cryosphere.**

**Main comments**

- **I think the discussion about the physical mechanisms that lead to reduced SMB on extreme negative SMB days ("SMB -ex") days needs to be sharpened and clarified. If I'm interpreting the results correctly, the study calculates changes in SMB on these days without considering whether melt is directly caused by rainfall or whether it's primarily the result of atmospheric warming that tends to occur on the same days as extreme precipitation. A full surface energy balance analysis that would be required to definitely answer this question is likely beyond the scope of this study. However, to avoid confusion I feel that this uncertainty about the physical drivers of melt should be addressed in more detail prior to the brief statement in L426–429. I note that prior studies (e.g. Doyle et al. 2015 ; Fausto et al. 2016a,b; Box et al. 2023) have found that the rain heat flux (i.e. the direct effect of rain on ice ablation) is relatively small (< 20%) even during extreme rainfall events, at least in the historical period.**
  We agree that a full surface energy balance analysis would be an excellent next step, but beyond the scope of the current work. A statement suggesting this future work has been added to Section 5.3, as it is a great recommendation to build upon the work presented here. (Lines 620-625):

*"Because of this, it is difficult to disentangle the effects of other climate variables from the effects of extreme precipitation. Indeed, the changes illustrated here are likely small contributors to the total decrease in SMB from melt due to rising temperatures but can still provide a better understanding of the processes impacting the SMB. Future work analysing the surface energy balance would allow for a more detailed understanding of the magnitude of the impacts associated with the precipitation itself versus other factors on extreme precipitation days."*

We have also added text to better outline the limitations of the analysis and be clear that we are not able to split the direct effects of the precipitation versus other effects. The following text has been added with the discussion of Figure 8 (lines 455-460): *"This analysis cannot quantify the extent to which this shift results specifically from the precipitation itself versus other factors, such as increased temperatures on extreme precipitation days. Historical case studies have estimated the direct effects of rainfall on ice to account for < 20% of total mass loss in studied events (e.g., Box et al., 2023; Doyle et al., 2015; Fausto et al., 2016a, b). While full surface energy balance analysis is required to assess the direct precipitation-related effects, changes, such as those shown in Figure 8, illustrate the potential that even days with the highest precipitation may not yield positive SMB in the future."*

A brief explanation of the mechanisms by which negative SMB may occur on extreme precipitation days has been added to the discussion of anomalies on $SMB_{ex}^-$ days in Figure 11 (lines 534-546): *"The anomalies illustrated in Fig. 11 show some of the mechanisms by which extreme precipitation days result in negative SMB. In western Greenland and Baffin Island, there are large increases in melt (Fig. 11e), which are collocated with reductions in albedo (Fig. 11d) and increased temperature (Fig. 11f). While we cannot quantify the drivers of the change in albedo, heavy rainfall may darken the surface and be a strong contributor to the negative albedo anomaly. These regions see a modest increase in refreezing (Fig. 11c), but it does not offset the increase in melt, leading to a large increase in runoff (Fig. 11b) and negative SMB anomaly (Fig. 11a)."*

- **It would be helpful to have more explanation about the somewhat counterintuitive changes in temperature anomalies during extreme SMB days in the future. Do the authors have an explanation for why the temperature anomalies during positive SMB JJAS extreme precipitation days are greater in the historical than the future simulation (Fig. 9–10, L357–360)? Is this because the temperature is already nearer to the freezing point in the future scenario and there is less overall variance in warm season temperature? And the finding of negative temperature anomalies in some low-lying and coastal areas on the future positive SMB JJAS extreme precipitation days (Fig. 10, L345–346) seems counterintuitive as well – do the authors have an explanation for this? To me it looks like the temperature anomalies are negative in the ablation zone and positive everywhere else on these days.**
We agree with your interpretations of the temperature anomalies in Figs. 9-11. Large temperature anomalies were historically associated with $SMB_{ex}^+$ historically because warmer air can hold more moisture to fuel intense precipitation. In a future warmer climate, less atmospheric temperature increase is required to yield intense extreme precipitation (i.e., similar to those obtained in the historical period), explaining the reduced positive temperature anomaly in Fig. 10. Some areas in the ablation zone show a negative future temperature anomaly associated with $SMB_{ex}^+$ as a negative anomaly is required to bring relatively warm summer air temperatures towards the freezing point, allowing for snowfall and favouring positive SMB anomalies.

Some text has been added to the manuscript to clarify these counterintuitive changes:

- Lines 478-481: *"Historically, the positive SMB extreme precipitation days generally occur with positive temperature anomalies (~3-4 K) and modest anomalies in melt, runoff, and albedo (Fig. 9). While positive temperature anomalies may usually contribute to melt, the warmer air can hold more moisture and feed heavy precipitation, which is likely to fall as snow in many high latitude/altitude regions during the warm season."*

- Lines 505-509: *"This reduction in temperature anomaly associated with $SMB_{ex}^{+}$ is likely due to the background increase in temperature, meaning the air can hold more moisture without requiring strong temperature anomalies. Some areas in the ablation zone show a negative future temperature anomaly associated with $SMB_{ex}^{+}$ as a negative anomaly is required to bring the relatively warm summer temperatures towards the freezing point, allowing for snowfall and favouring positive SMB anomalies."*

- **In addition to the increase in atmospheric moisture with warming explained by the Clausius-Clapeyron relationship, is it possible that the increased contribution of extreme precipitation days to positive SMB in northern Greenland and northern Ellesmere Island has some contribution from Arctic sea ice decline? See L237–241, L305–307, L470–474.**
  Yes, it is very possible that the increased contribution of extreme precipitation days to positive SMB in the northern areas of the domain is at least partially due to the decline in Arctic sea ice. While it is outside of the scope of this study to prove whether or not this is true, it is clear that the loss of sea ice leads to vast increases in local evaporation in the Arctic (e.g., Bintanja et al., 2014; Kopec et al., 2016), as well as potentially enhancing baroclinicity and lower static stability (e.g., Crawford et al., 2022; Koyama & Stroeve, 2017), which may accelerate storm intensification and therefore enhance the precipitation rates.

  Kopec et al. (2016) estimated that precipitation increases by 21.1$\pm$ 9.1% per $10^5$ km$^2$ of Arctic sea ice lost at 6 sites in the Arctic (including within our study region). Similarly, Hartmuth et al. (2023) showed that projected precipitation increases in the Arctic are largest in regions with ongoing ice retreat.

  However, we cannot explicitly test this as the VR-CESM simulations have monthly prescribed sea ice conditions (as they are partially-coupled Atmospheric Model Intercomparison Project-style simulations) and the forcing data for RACMO is no longer accessible. While we can't investigate this in detail, we do expect that if the data were available and the simulations contained a coupled sea ice model, we would see this type of relationship.

  Some text has been added to reference the potential importance of sea ice loss to this change:
  - Section 4.1 (lines 322-326): "VR-CESM shows some general SMB increases on extreme precipitation days, particularly in NO Greenland. This is likely due to the increase in the magnitude of extreme precipitation events, as warmer air can hold more moisture (e.g., Bengtsson et al., 2011; Norris et al., 2019; Skific et al., 2009)*, which may be further enhanced by the loss of Arctic sea ice (e.g., Bintanja and Selten, 2014; Hartmuth et al., 2023; Kopec et al., 2016)"*

  - Section 6 (lines 683-687): "Future changes are generally smaller in the cold season, when the most notable change is a decrease in the contribution of extreme precipitation days to positive SMB in SE Greenland. Small increases across the northernmost regions of the domain reflect the increased water vapour holding capacity of warmer air, which allows for more cold season extreme precipitation, *and may also be facilitated by sea ice loss and enhanced moisture*

*availability."*

References:

Bintanja, R., and F. M. Selten, 2014: Future increases in Arctic precipitation linked to local evaporation and sea-ice retreat. *Nature*, **509**, 479–482, https://doi.org/10.1038/nature13259.

Crawford, A. D., J. V. Lukovich, M. R. McCrystall, J. C. Stroeve, and D. G. Barber, 2022: Reduced Sea Ice Enhances Intensification of Winter Storms over the Arctic Ocean. J Climate, 35, 3353–3370, https://doi.org/10.1175/jcli-d-21-0747.1.

Hartmuth, K., Papritz, L., Boettcher, M., & Wernli, H. (2023). Arctic Seasonal Variability and Extremes, and the Role of Weather Systems in a Changing Climate. Geophysical Research Letters, **50**(8), e2022GL102349. https://doi.org/10.1029/2022GL102349

Kopec, B. G., X. Feng, F. A. Michel, and E. S. Posmentier, 2016: Influence of sea ice on Arctic precipitation. Proc. Natl. Acad. Sci., **113**, 46–51, https://doi.org/10.1073/pnas.1504633113.

Koyama, T., J. Stroeve, J. J. Cassano, and A. D. Crawford, 2017: Sea ice loss and Arctic cyclone activity from 1979 to 2014. *Journal of Climate*, **30**, 4735–4754, https://doi.org/10.1175/jcli-d-16-05https://doi.org/10.1175/jcli-d-16-0542.1.

- **Greenland Ice Sheet, the study areas merge elevation zones of the ice sheet that have distinct characteristics in the current climate, from the low-elevation ablation zone up through the percolation zone and high-elevation accumulation zone. I think it would be a really helpful addition to the paper to provide a few analyses of how these results vary with elevation in the past and future simulations. For example, the warm season panels of Fig. 2 (described in L225–226) imply a huge expansion of the annual ablation zone into higher elevations, and it would be interesting to know the elevation range of the historical and future ablation zone. In Fig. 8 (L325–326), it would be interesting to know the elevation ranges of the areas over which extreme precipitation days make substantial negative contributions to JJAS SMB in the historical and future simulations.**
Excellent point – we have made some edits to highlight changes in the ablation zone, including adding a line denoting the mean ablation zone extent for each simulation in the map figures to facilitate the identification of changes within the ablation zone and the ablation zone expansion.

Some discussion has been added about the expansion of the ablation zone (and difference between the two models) Section 4.1 (lines 285-299): "*RACMO also shows strong decreases in SMB reaching further inland than VR-CESM. Both models showed the ablation zone similar altitudes historically (1742 m and 1830 m in RACMO and VR-CESM, respectively), though the ablation zone in RACMO covers ~17.7% of the GrIS compared to only 9.4% in VR-CESM. However, in the future, both models show the ablation zone expanding to cover an additional ~28% of GrIS area. RACMO shows expansion of the ablation zone to altitudes up to 2658 m, compared to only 2297 m in VR-CESM. These differences between RACMO and VR-CESM are consistent with those found by van Kampenhout et al. (2019), which also showed the largest differences in the ablation zone with VR-CESM producing higher SMB than RACMO.*"

**Other comments**

- **L1: I suggest changing the title to incorporate the fact that the impacts of both historical and future extreme precipitation events are studied in this paper, and/or to emphasize that the key takeaway is that the impacts of extreme precipitation events are expected to qualitatively change in the future. Maybe "Modeling the impacts of extreme historical and future precipitation events...", or "Changes in future impacts of extreme precipitation events on surface mass balance in the eastern Canadian Arctic and Greenland".**
  Great suggestion – we have edited the title to "Modelling the Impacts of Historical and Future Extreme Precipitation Days on Seasonal Surface Mass Balance in the Eastern Canadian Arctic and Greenland" to more accurately reflect the material in the paper.

- **L28–39: I suggest revising the topic sentence of this paragraph to state that Arctic land ice is inclusive of both the Greenland Ice Sheet and the ice caps and glaciers of the eastern Canadian Arctic. As written, the eastern Canadian Arctic Archipelago is introduced rather abruptly near the end of the paragraph.**
  The first sentence of the paragraph has been edited to highlight the study region earlier – Lines 28-29: "Arctic land ice, *including the Greenland Ice Sheet (GrIS) and glacier and ice caps of the eastern Canadian Arctic*, has been losing mass at an accelerated rate as the climate has warmed (e.g., Hugonnet et al. 2021; Constable et al. 2022)."

- **L46: Be specific that increased water vapour holding capacity is due to climate warming**
  The sentence has been edited to clarify this – Lines 55-57: *"In general, precipitation is expected to increase in most glaciated regions due to increased water vapour holding capacity as a result of atmospheric warming (e.g., Bengtsson et al., 2011; Norris et al., 2019; Skific et al., 2009)"*

- **L53–62: Nice explanation of potential changes in firn structure and response to precipitation due to climate warming.**
  **Thank you!**

- **L150–152: I like the idea to use four-month seasons (JJAS warm season, DJFM cold season)**
  Thank you!

- **L196–198: The result that VR-CESM exhibits little change in extreme precipitation SE Greenland in any month appears to be consistent with the previous study by this lead author (Loeb et al., 2024), which found that VR-CESM projects decreases in extreme precipitation in SE Greenland. However this study shows that the RACMO-downscaled CESM simulation projects an extreme in warm season precipitation. Does this suggest that the results of the prior study were specific to the VR-CESM model?**
  Thank you for highlighting this – a small error was found upon looking into the calculation for RACMO's mean monthly extreme precipitation, and with the correct calculation, RACMO and VR-CESM do agree more closely (both in general and on the changes in SE Greenland specifically). Apologies for this error! The largest changes from the original Figure 2 to the updated version are: SE Greenland (now showing a decrease in winter and small increase in summer), Baffin Island (pattern now aligns closely with VR-CESM), and Devon Island (much larger increase in summer extreme precipitation).

[Figure]

- **Figs. 9–11 and associated text: I'm a little confused about the relationship between modeled melt, runoff, and refreezing in these results. For example, does Fig. 10 show that on future positive SMB JJAS extreme precipitation days, there is anomalously large amounts of melt along the margins of the Greenland Ice Sheet, but less runoff and refreezing in most of these same areas? Or does this show that the amount of meltwater produced is below normal on those days? And if the latter, how is there anomalously large runoff in the lower elevations of SE Greenland despite there being less melt than normal? It would be helpful to give more detail about the physical processes represented by each term and the relationship between them, including whether a positive/negative value of each variable represents a positive or negative contribution to SMB.**

In Figure 10e, the melt anomalies along the periphery are negative, meaning that the melt occurring on the extreme precipitation is less than that of the surrounding time period and the refreezing anomaly along most low-lying areas is positive as well. In terms of the collocated positive runoff anomaly, we believe this is likely due to mixed precipitation falling – we've added new supplementary figures (S7-8) that show the mean extreme precipitation amounts and rain fraction for positive and negative events in each time period, and in the future positive events the rain fraction looks to be ~0.5 or just above in some of the areas in SE Greenland in RACMO (slightly lower rain fraction in VR-CESM). This would help explain the positive SMB anomaly while still increasing the runoff.

Note that the colorbars for each panel are oriented such that the blue colours suggest anomalies that act to increase the SMB (e.g., blue on panel B indicates reduced runoff, and blue on panel C indicates increased refreezing). The following sentence has been added to the caption for Figure 9 to clarify this: "*Blue colours in each panel indicate anomalies that act to increase SMB.*"

**Technical corrections**

- **L22: Add comma after "future"**
  Comma has been added

- **L67: The word "cause" is grammatically incorrect here. Should this be "causing" or "and cause"?**
  Edited to "causing"

- **L288: Typo with "Figure 6" inserted at the beginning of a sentence with no space. Please check this elsewhere, e.g. L329, 335–336**
  Corrected

**References**

Box, J. E., Nielsen, K. P., Yang, X., Niwano, M., Wehrlé, A., Van As, D., Fettweis, X., Køltzow, M. A. Ø., Palmason, B., Fausto, R. S., Van Den Broeke, M. R., Huai, B., Ahlstrøm, A. P., Langley, K., Dachauer, A., & Noël, B. (2023). Greenland ice sheet rainfall climatology, extremes and atmospheric river rapids. _Meteorological Applications_, _30_(4), e2134. https://doi.org/10.1002/met.2134

Doyle, S. H., Hubbard, A., van de Wal, R. S. W., Box, J. E., van As, D., Scharrer, K., Meierbachtol, T. W., Smeets, P. C. J. P., Harper, J. T., Johansson, E., Mottram, R. H., Mikkelsen, A. B., Wilhelms, F., Patton, H., Christoffersen, P., & Hubbard, B. (2015). Amplified melt and flow of the Greenland ice sheet driven by late-summer cyclonic rainfall. _Nature Geoscience_, _8_(8), 647–653. https://doi.org/10.1038/ngeo2482

Fausto, R. S., As, D., Box, J. E., Colgan, W., Langen, P. L., & Mottram, R. H. (2016). The implication of nonradiative energy fluxes dominating Greenland ice sheet exceptional ablation area surface melt in 2012. _Geophysical Research Letters_, _43_(6), 2649–2658. https://doi.org/10.1002/2016GL067720

Fausto, R. S., van As, D., Box, J. E., Colgan, W., & Langen, P. L. (2016). Quantifying the Surface Energy Fluxes in South Greenland during the 2012 High Melt Episodes Using In-situ Observations. _Frontiers in Earth Science_, _4_. https://doi.org/10.3389/feart.2016.00082

---

## Author Response (AR2)

**Author's Response for Modelling the Impacts of Historical and Future Extreme Precipitation Days on Seasonal Surface Mass Balance in the Eastern Canadian Arctic and Greenland**

Thank you to the editor and anonymous reviewers for their feedback during the review process and support in publishing the manuscript. The files uploaded match the previous version, except that the tick labels for the colourbars for panels (c) and (d) of figures 9-11 have been adjusted slightly to align with the breaks in the colormap used in the figure.